# Management and Outcome of Young Women (≤40 Years) with Breast Cancer in Switzerland

**DOI:** 10.3390/cancers14051328

**Published:** 2022-03-04

**Authors:** Giacomo Montagna, Robin Schaffar, Andrea Bordoni, Alessandra Spitale, Daniela A. Terribile, Lorenzo Rossi, Yvan Bergeron, Bernadette W. A. van der Linden, Isabelle Konzelmann, Sabine Rohrmann, Katharina Staehelin, Manuela Maspoli-Conconi, Jean-Luc Bulliard, Francesco Meani, Olivia Pagani, Elisabetta Rapiti

**Affiliations:** 1Breast Service, Department of Surgery, Memorial Sloan Kettering Cancer Center, New York, NY 10001, USA; montagng@mskcc.org; 2Geneva Cancer Registry, University of Geneva, 1205 Geneva, Switzerland; robin.schaffar@unige.ch; 3Ticino Cancer Registry, Cantonal Institute of Pathology, 6600 Locarno, Switzerland; andrea.bordoni@ti.ch (A.B.); alessandra.spitale@ti.ch (A.S.); 4Dipartimento di Scienze Mediche e Chirurgiche, Università Cattolica del Sacro Cuore, 00168 Rome, Italy; daniterribile@gmail.com; 5Multidisciplinary Breast Centre, Dipartimento Scienze della Salute della Donna e del Bambino e di Sanità Pubblica, Fondazione Policlinico Universitario A. Gemelli IRCCS, 00168 Rome, Italy; 6Oncology Institute of Southern Switzerland, 6500 Bellinzona, Switzerland; lorenzo.rossi@eoc.ch; 7Breast Unit of Southern Switzerland, Ente Ospedaliero Cantonale, 6900 Lugano, Switzerland; francesco.meani@eoc.ch (F.M.); opagani@bluewin.ch (O.P.); 8Fribourg Cancer Registry, 1705 Fribourg, Switzerland; yvan.bergeron@liguassante-fr.ch (Y.B.); bernadette.vanderlinden@unifr.ch (B.W.A.v.d.L.); 9Population Health Laboratory (#PopHealthLab), University of Fribourg, 1705 Fribourg, Switzerland; 10Valais Health Observatory (OVS), 1950 Sion, Switzerland; isabelle.konzelmann@ovs.ch; 11Cancer Registry Zurich, Zug, Schaffhausen and Schwyz, University Hospital Zurich, 8091 Zurich, Switzerland; sabine.rohrmann@usz.ch; 12Basel Cancer Registry, 4001 Basel, Switzerland; katharina.staehelin@usb.ch; 13Neuchâtel and Jura Cancer Registry, 2000 Neuchâtel, Switzerland; manuela.maspoli@ne.ch (M.M.-C.); jean-luc.bulliard@unisante.ch (J.-L.B.); 14Vaud Cancer Registry, Centre for Primary Care and Public Health (Unisanté), University of Lausanne, 1010 Lausanne, Switzerland; 15Department of Obstetrics and Gynecology, Ente Ospedaliero Cantonale (EOC), 6900 Lugano, Switzerland; 16Geneva University Hospitals, 1205 Geneva, Switzerland; 17Swiss Group for Clinical Cancer Research (SAKK), 6500 Bellinzona, Switzerland

**Keywords:** young women, breast cancer, Switzerland, management, outcome

## Abstract

**Simple Summary:**

Data on the outcome and treatment of young women with breast cancer (BC) in Switzerland is scarce. We conducted a retrospective cohort study to evaluate treatment and outcome of women aged ≤ 40 years, diagnosed with stage I-III BC in Switzerland between 2000–2014. We found that the majority of patients were treated according to international guidelines, however we identified differences in quality-of-care score across the two Swiss linguistic/geographic regions (Swiss Latin and Swiss German). Survival was high: 91.4% (95% confidence interval (CI) 90.2–92.5) at 5 years and 83.1% (95% CI 81.2–78.5) at 10 years. After adjusting for multiple clinicopathological factors only tumor characteristics and treatment period remained independently associated with survival. We concluded that national guidelines for young women with BC should be implemented to standardize treatment in Switzerland and awareness should be raised among young women and clinicians that BC does not discriminate by age.

**Abstract:**

Background: An increase in breast cancer (BC) incidence in young women (YW) as well as disparities in BC outcomes have been reported in Switzerland. We sought to evaluate treatment and outcome differences among YW with BC (YWBC). Methods: YW diagnosed with stage I-III BC between 2000–2014 were identified through nine cancer registries. Concordance with international guidelines was assessed for 12 items covering clinical/surgical management, combined in a quality-of-care score. We compared score and survival outcome between the two linguistic-geographic regions of Switzerland (Swiss-Latin and Swiss-German) and evaluated the impact of quality-of-care on survival. Results: A total of 2477 women were included. The median age was 37.3 years (IQR 34.0–39.4 years), with 50.3% having stage II BC and 70.3% having estrogen receptor positive tumors. The mean quality-of-care score was higher in the Latin region compared to the German region (86.0% vs. 83.2%, *p* < 0.0005). Similarly, 5- and 10-year overall survival rates were higher in the Latin compared to the German region (92.3% vs. 90.2%, *p* = 0.0593, and 84.3% vs. 81.5%, *p* = 0.0025, respectively). There was no difference in survival according to the score. In the univariate analysis, women in the Latin region had a 28% lower mortality risk compared to women in the German region (hazard ratio 0.72; 95% CI 0.59–0.89). In the multivariable analysis, only stage, differentiation, tumor subtype and treatment period remained independently associated with survival. Conclusions: We identified geographic disparities in the treatment and outcome of YWBC in Switzerland. National guidelines for YWBC should be implemented to standardize treatment. Awareness should be raised among YW and clinicians that BC does not discriminate by age.

## 1. Introduction

Breast cancer (BC) survival in Switzerland is high (with relative survival rates of 87.9% (95% CI 87.3–88.5) at 5 years and 80.1% (95% CI 79.2–81.0) at 10 years) [1]. Breast Cancer in Young Women (YW), defined as BC diagnosed in women aged ≤40 years, is rare and represents approximately 6% of all BC cases in developed countries [2]. However, an increase in BC in YW was reported in several countries, including Switzerland [3,4,5]. Because BC screening is not recommended for this age group, YW are more likely to be diagnosed with symptomatic disease and at a more advanced stage [5,6]. Additionally, YW have a greater proportion of triple-negative and human epidermal growth factor 2 (HER2)-positive tumors and worse outcomes than older women, especially for luminal tumors [7]. Evidence-based international guidelines to optimize treatment, minimize side effects, and improve outcomes of YW with BC (YWBC) have been specifically developed [8,9,10,11].

The Swiss healthcare system is characterized by universal coverage ensured through mandatory health insurance, readily available access, and care provision decentralization and fragmentation. The latter aspect is attributable to the fact that healthcare is mostly the responsibility of the 26 cantons and, thus, varies among them [12]. Previous studies described geographic disparities in BC early detection and treatment across the country with significant variability in the type of axillary staging procedures, reconstruction rates, and recommendations for the use of endocrine- and chemotherapy [13]. These aspects and differences in treatment attitudes can partly explain the regional disparities in the BC survival and mortality rates in Switzerland [14,15].

To date, no comprehensive information exists at the national level on the management, treatment, and outcomes of YWBC in Switzerland. We sought to evaluate quality-of-care and survival in YWBC in the two linguistic-geographic regions of the country (Latin and German).

## 2. Methods

### 2.1. Study Population and Data Source

Women aged ≤ 40 years diagnosed with primary invasive BC between 2000 and 2014, were identified through nine population-based cancer registries (Basel (Stadt and Landschaft), Fribourg, Geneva, Neuchatel, Jura, Vaud, Ticino, Wallis, and Zurich), covering approximately 45% of the Swiss population and including four out of five Swiss University Hospitals (Figure 1). All registries included cases diagnosed during 2000–2014 except the registries of Basel (2000–2011), Fribourg (2006–2014), and Jura (2005–2014) (Appendix A).

Patients meeting the following criteria were excluded: diagnosis made at autopsy, non-epithelial BC, and case information restricted to death certificate.

### 2.2. Variables

Variables of interest routinely collected by the cancer registries included age, period (2000–2004, 2005–2009, 2010–2014, missing), methods of diagnosis (mammography screening, clinical screening, breast self-examination, other (including symptoms or incidental finding), or unknown), tumor stage classified according to the Tumor Node Metastasis (TNM) pathological system (0, I, II, III, IV, unknown), tumor differentiation (well, moderately, or poorly differentiated, or unknown), tumor histology (ductal, lobular, or other), estrogen and progesterone receptor status (positive if ≥1% expressed, negative, or unknown), locoregional (including type of breast surgery received (none, lumpectomy, or mastectomy) and type of axillary surgery (none, sentinel lymph node biopsy or axillary lymph node dissection)), and systemic treatment (receipt of chemotherapy, endocrine therapy and target therapy).

An ad hoc questionnaire for variables not routinely recorded by the registries (including family history (FH) of BC, sector of care, multidisciplinary discussion, and markers not routinely recorded from the beginning of the study period (i.e., Ki-67 and HER2 status)) was developed and filled out by trained registrars in each registry after reviewing medical charts. Tumor subtypes were defined as follows:Triple-negative: estrogen receptor (ER) and progesterone receptor (PR) staining <1% and HER2 immunohistochemical score classified as 0 or 1+. In case of equivocal HER2 status (2+), tumors were considered HER2− if fluorescence in situ hybridization (FISH) showed an HER2-to-probe ratio < 2.0.HER2+: ER/PR < 1% and HER2 immunohistochemical score of 3+ or HER2-to-probe FISH ratio ≥ 2.0.Luminal A-like: ER/PR > 1% HER2- and Ki-67 < 14% or ER/PR > 1% HER2- and well or moderately differentiated tumors.Luminal-B-like: ER/PR > 1% HER2+ or ER/PR > 1% HER2- and Ki-67 ≥ 14%, or ER/PR > 1% HER2- and poorly differentiated tumors.

A positive FH was defined as any first- or second-degree relative affected by BC.

### 2.3. Covariate Measures

Based on the aims of the study and on clinical knowledge, the following covariates were considered in the survival model: age (categorized in 4 groups (<25, 25–29, 30–34, 35–40 years) for description and recategorized as (<35, >35 years) in the models), period of diagnosis, tumor differentiation, tumor stage and subtype, linguistic-geographic region, and quality-of-care score.

### 2.4. Linguistic-Geographic Region

The linguistic-geographic regions were created by dividing the cantons into two groups representing the two main Swiss language regions (Basel and Zurich in the German region and all the remaining ones in the Latin (French/Italian) region), as previous reports have shown differences in cancer risk avoidance behaviors, use of health care services, and socioeconomic status in the two regions [13,16]. The attribution to the linguistic-geographic region was determined based on the residence of the patient at the time of diagnosis and it was considered to be unrelated to the place of birth or nationality.

### 2.5. Quality-of-Care Score

Concordance with international guidelines available at the time of diagnosis including the European Society of Breast Cancer Specialists (EUSOMA) recommendations [17,18], St. Gallen International Breast Cancer Consensus Guidelines [19,20,21,22,23,24,25], and ESO-ESMO International Consensus Guidelines for Breast Cancer in Young Women (BCY) [8] was assessed for 12 items covering clinical and surgical management combined in a quality-of-care score. The following indicators were assessed:For clinical management: method of diagnosis, time to start of treatment of <6 weeks from diagnosis, complete pathology report (pT, pN, tumor grade, ER%, and PR%);For locoregional treatment: negative final margins, 1 surgical procedure only, ≥10 lymph nodes removed if axillary lymph node dissection (ALND) performed, radiotherapy (RT) following breast-conserving surgery (BCS), RT boost if BCS, and RT after mastectomy when indicated;For systemic therapy: receipt of (neo)adjuvant chemotherapy, receipt of (neo)adjuvant anti-HER2 therapy, and prescription of endocrine therapy.

Indicators were scored as “1” when correctly performed, “0” when not correctly performed, and “not applicable” when the therapy was not applicable to the type or stage of cancer. We calculated an overall quality-of-care score for each patient. Missing datapoints were excluded from the quality-of-care score calculation. The overall score for a given patient was the ratio of the sum of all scored indicators by the number of indicators applicable for that patient.

We defined quality of care according to the distribution of the score. We categorized the quality of care into 4 categories: patients with a score of 100% were included in one group; the others were divided in tertiles: high-, medium-, and poor-quality care.

### 2.6. Outcome Measures

Survival time was the outcome of interest. We defined it as the time between the date of diagnosis and the date of death for any cause, the date of departure from the canton, or the end of follow-up (31 December 2018), whichever came first. Cancer-specific survival was estimated using net survival in the relative survival setting. Net survival can be interpreted as the probability of surviving BC in the absence of any other cause of death, by accounting for potential differences in other-cause mortality rates across Swiss cantons.

### 2.7. Statistical Analysis

Patient and tumor characteristics, treatment setting, and quality-of-care scores were compared between the two linguistic-geographic regions using a *t*-test or chi-square test.

Two measures were used to evaluate survival. First, overall survival considered death from all causes. Kaplan–Meier estimates for overall survival (OS), by region and by score, were compared using log-rank test. Then, univariate and multivariable time-to-event analyses (Cox regression model for OS and Royston–Parmar [26] model for net survival) were conducted to evaluate the impact of covariables on survival. All covariates that were significant at the α < 0.05 level in the univariate analysis were considered in the multivariable model. A separate category for missing cases for each variable was included in the models. All tests were evaluated for statistical significance at alpha level 0.05. Statistical analysis was performed using STATA (Version 15.1, StataCorp, College Station, TX, USA).

Due to missingness, we performed two sensitivity analyses considering extreme case scenarios. In the first scenario, we attributed a score of “1” to all applicable but missing cases as if they were performed correctly. In the second analysis, we considered that all applicable but missing cases were not correctly performed and received a score of “0”.

### 2.8. Ethics

The study was approved by the ethical committees of the Geneva and Zurich Cantons (no. 2017-01074).

## 3. Results

We included 2477 women, 1469 of which were from the Latin region and 1008 of which were from the German region. The median age was 37.3 years (IQR 34.0–39.4) for both regions. The majority of patients had stage II, poorly differentiated, ER+ tumors, and were treated in the public sector. Women in the Latin region were more likely to be diagnosed with stage I disease (36.4% vs. 30.5%, *p* = 0.013) and with Luminal-B-like tumors (50.4% vs. 43.9%, *p* = 0.034) and to be diagnosed between 2010 and 2014 (38% vs. 31.9%, *p* = 0.006), but they were less likely to have a positive FH (37.7% vs. 44.9%, *p* = 0.005) (Table 1).

### 3.1. Quality-of-Care Score

Table 2 shows the scores for each individual quality-of-care indicator according to the linguistic-geographic region. Differences between the two regions were in the proportion of women with ≥10 lymph nodes removed at ALND (75.8% in the Latin region vs. 82.5% in the German region, *p* = 0.002) and of women receiving whole breast RT and boost after BCS (92.5% and 95.8% in the Latin region vs. 75.0% and 88.6% in the German region, respectively, both *p* < 0.0001). Significant differences were also seen in the prescription and receipt of systemic therapy including adjuvant chemotherapy, anti-HER2 therapy and endocrine therapy, being more often used in the Latin region. However, the high proportion of missing datapoints for anti-HER2 therapy made data comparison difficult.

Overall, the mean score was 86.0 (standard deviation (SD) 15.8) for the Latin region and 83.2 (SD 16.5) for the German region (*p* < 0.0005) with more than 50% of women in the Latin region having a score >88 compared to only 28% in the German region (Table 3). To confirm these findings, we performed two sensitivity analyses. In the first, we attributed a score of “1” to all applicable but missing cases in both regions as if they were performed correctly. The mean quality-of-care score in the Latin region remained higher than in the German region (87.6 (SD 13.0) vs. 85.0 (SD 14.1), *p* < 0.0005). In the second analysis, we considered that all applicable but missing cases were not correctly performed and received a score of “0”. Again, the mean quality-of-care score in the Latin region remained higher than in the German region (82.5 (SD 18.8) vs. 78.4 (SD 19.1), *p* < 0.0005).

### 3.2. Survival

The median follow-up for the entire cohort was 7.6 years. The OS in the full cohort was 91.4% (95% confidence interval (CI) 90.2–92.5) at 5 years and 83.1% (95% CI 81.2–78.5) at 10 years, and was higher for women in the Latin region compared to those in the German region (5-year OS: 92.3% (95% CI 90.1–93.6) vs. 90.2% (95% CI 88.0–91.9), *p* = 0.0593; 10-year OS: 84.3% (95% CI 81.8–86.4) vs. 81.5% (95% CI 78.4–84.2), *p* = 0.0025, respectively) (Figure 2). In the univariate analysis, there was no difference in the OS according to quality-of-care score tertiles (Appendix A). In the multivariable analysis, the linguistic-geographic region was not independently associated with the OS (Latin region (referent): HR 0.86, 95% CI 0.70–1.07). Factors that were independently associated were tumor stage and differentiation, tumor subtype, and period of diagnosis (Table 4). The multivariable analysis for net cancer-specific survival showed similar results (Appendix A).

## 4. Discussion

In this study, we evaluated compliance with international guidelines for the management of YWBC (≤40 years) and survival in the two linguistic-geographic regions of Switzerland between 2000 and 2014. Overall, the great majority of YWBC received guidelines-concordant care, with women treated in the Latin region having a significantly higher score and higher OS compared to those treated in the German region. However, after adjusting for clinicopathological factors and period of diagnosis, the OS was not significantly associated with the quality-of-care score or the linguistic-geographic region.

The finding of high compliance with guidelines-recommended treatment in Switzerland mimics the results of a recent study from the United States. The authors analyzed data from 952 YWBC diagnosed in 2013 in the National Cancer Institute’s Patterns of Care Study and found that 81.7% received guideline-concordant care [27]. This is promising as adherence to treatment guidelines is associated with improved BC outcomes [28] and shows that physicians are willing to follow evidence-based guidelines for this specific group of patients.

Notably, the survival rate of our population was very high (91.4% and 83.1% at 5 and 10 years, respectively) compared to other studies conducted in this age group. In the POSH study, which evaluated the effect of germline BRCA1/2 mutation on BC survival in YW diagnosed in the United Kingdom between 2000 and 2008, the 5-year OS was 83.8% among mutation carriers and 85.0% among noncarriers; the 10-year OSs were 73.4% and 70.1%, respectively [29]. In a study from the Memorial Sloan Kettering Cancer Center including 529 very YW (≤35 years) with stage I-III BC treated between 1990 and 2010, only 73% of patients were alive at 10 years [30]. However, the high OS rates in our study could also be explained by the high proportion of patients (over two-thirds) having hormone receptor positive tumors, known to have better outcomes compared to other tumor subtypes [31].

Geographic heterogeneity in the early detection and treatment of BC patients in Switzerland was previously reported. Ess et al. analyzed data of nearly 5000 women diagnosed with early BC between 2003 and 2005 and found considerable geographic variation in the proportion of women with early diagnosis, mastectomies and reconstruction rate, use of sentinel lymph node biopsy, and compliance with recommendations on the use of endocrine therapy and chemotherapy [13].

However, we did not find an association between the quality-of-care score and OS. There are multiple possible explanations for such a result. First, the mean quality-of-care score was very high across the entire cohort, with only 2.7% (67/2477) of women having a score <50%. This may have limited our ability to detect the potential association between a low score and poor survival. Second, the indicators we used, although important to measure adherence to guidelines, are not reliable predictors of a particular outcome. None of the variables we included are specific to YWBC. Lastly, the high rate of missing data for some of the indicators may also explain the lack of association.

Previous studies have found significant differences in BC survival across Swiss regions; Fisch et al. analyzed data from over 11,000 patients diagnosed with BC between 1988 and 1997 and found regional OS differences after adjustment for age and stage. In their study, survival was lowest in the rural parts of German-speaking regions (eastern Switzerland) and highest in urbanized regions of the Latin- and German-speaking regions (northwestern Switzerland) [14]. In the present study, we confirmed regional differences in outcomes with 5- and 10-year OS being higher in the Latin region compared to the German region. However, this difference did not persist after adjustment for other clinicopathological factors, and only tumor stage, grade, tumor subtype, and treatment period remained independently associated with survival. Given the prognostic role of tumor stage at diagnosis and that screening programs are not recommended for this age group, efforts should be made to raise awareness among YW and physicians. This is particularly important because while some authors have reported that most YW do not experience delays after breast abnormality detection [32,33], others found that many YW diagnosed with locally advanced BC experienced doctor delays [33].

The main limitations of the study are related to its retrospective nature. This explains the high number of missing data for variables not routinely collected by cancer registries and the number and type of variables included in the quality-of-care score. In addition, we were not able to capture other potential confounders such as socioeconomic and racial/ethnical factors, although the universal health insurance coverage and the racial homogeneity of the Swiss population make the confounding from these factors less likely. We were able to include only nine registries covering ~45% of the Swiss population because most of the other Swiss cancer registries did not cover the study period. This could potentially explain the baseline difference in tumor characteristics between the two regions. Additionally, data on treatment after the first 6–12 months from diagnosis and locoregional and distant recurrences were not available, affecting the accuracy of some indicators and the ability to provide patterns of disease recurrence. The strengths of the study are the large sample size of YWBC under study and the long follow-up, the inclusion of women from both linguistic-geographic regions and from the public and private sectors.

## 5. Conclusions

This was the first study evaluating concordance with international guidelines and outcomes of YWBC in Switzerland. In the setting of a healthcare system characterized by high expenditures, universal access to services, and high decentralization, we found that guideline concordance was high and survival rates were very high. However, the level of care varied according to the geographic region and disparities can be reduced with implementation of national guidelines for the care of YWBC. After adjustment for other clinicopathological factors, OS was not affected by the linguistic-geographic region, but by cancer features at diagnosis such as stage, grade, and subtype. Given that mammographic screening is not recommended for YW, until other screening methods are recognized as effective, our results highlight the importance of increasing BC awareness among YW. Additionally, physicians should be aware that BC does not discriminate by age and particular attention should be paid to FH and early sign and symptoms that can be easily mistaken as benign findings.

## Figures and Tables

**Figure 1 cancers-14-01328-f001:**
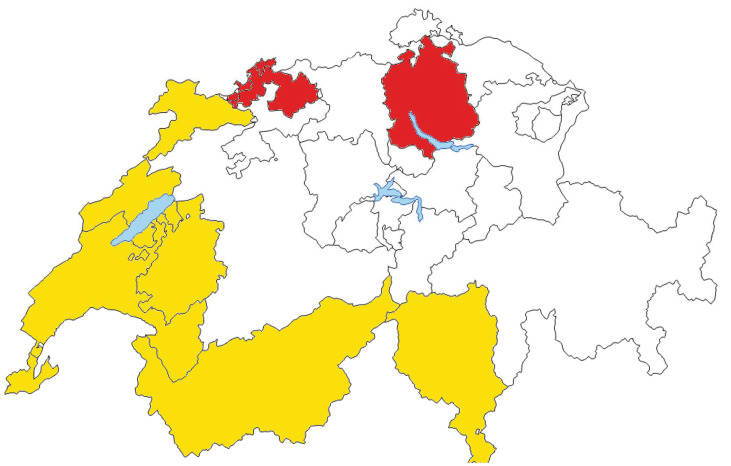
**Swiss regions (cantons) participating in the study.** Swiss-Latin cantons, including Swiss-Italian and Swiss-French, are depicted in yellow. Swiss-German cantons are depicted in red.

**Figure 2 cancers-14-01328-f002:**
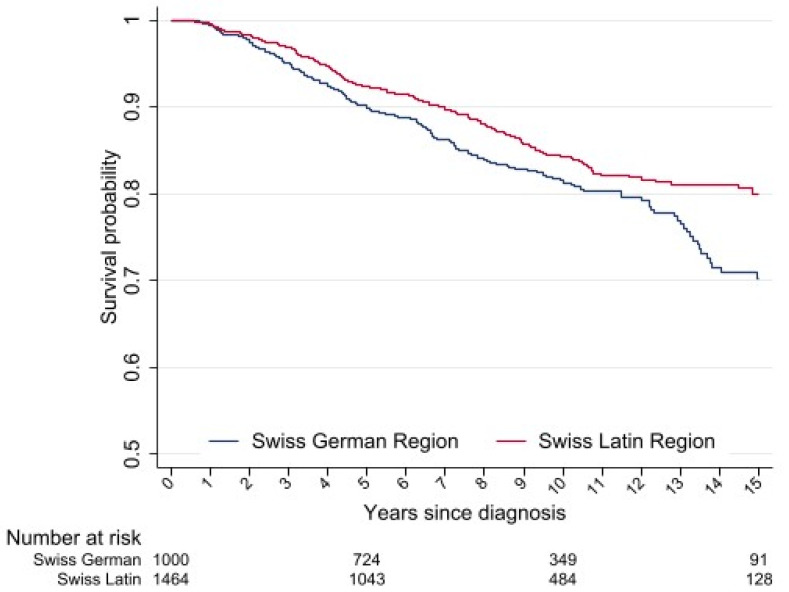
Kaplan–Meier estimates for overall survival of young women with breast cancer treated between 2000 and 2014 in Switzerland.

**Table 1 cancers-14-01328-t001:** Clinicopathological characteristics of the study cohort, stratified by linguistic-geographic region.

Characteristics	Total *n* = 2477	Latin *n* = 1469	German *n* = 1008	*p*-Value
**Median, years (IQR)**	37.3 (34.0–39.4)	37.3 (34.1–39.4)	37.3 (33.9–39.5)	0.880
**Age, *n* (%)**				0.811
<25	22 (0.9)	15 (1.0)	7 (0.7)	
25–29	160 (6.5)	93 (6.3)	67 (6.6)	
30–34	584 (23.6)	338 (23.0)	246 (24.4)	
35–40	1709 (69)	1021 (69.6)	688 (68.3)	
Unknown	2	2	0	
**Period of Diagnosis**				**0.006**
2000–2004	731 (29.5)	409 (27.9)	322 (31.9)	
2005–2009	864 (34.9)	500 (34.1)	364 (36.1)	
2010–2014	880 (35.6)	558 (38.0)	322 (31.9)	
Missing	2	2	0	
**Stage**				**0.013**
I	789 (34.1)	507 (36.5)	282 (30.5)	
II	1165 (50.3)	673 (48.3)	492 (53.2)	
III	362 (15.6)	212 (15.2)	150 (16.2)	
Unknown	161	77	84	
**Tumor differentiation**				**<0.0001**
Well	207 (8.7)	143 (10.1)	64 (6.5)	
Moderately	1019 (42.6)	630 (44.6)	389 (39.7)	
Poorly	1166 (48.7)	639 (45.3)	527 (53.8)	
Unknown	85	57	28	
**Tumor subtype**				**0.034**
Triple negative	432 (20.3)	244 (19.1)	188 (22.0)	
HER2+	150 (7.0)	86 (6.7)	64 (7.5)	
Luminal B-like	1019 (47.8)	644 (50.4)	375 (43.9)	
Luminal A-like	531 (24.9)	304 (23.8)	227 (26.6)	
Unknown *	345	191	154	
**Family history**				**0.005**
Negative	1058 (60.2)	768 (62.3)	290 (55.1)	
Positive **	700 (39.8)	464 (37.7)	236 (44.9)	
Unknown	719	237	482	
**Treatment setting**				**<0.0001**
Public	1238 (69.4)	809 (68.0)	429 (72.2)	
Private	546 (30.6)	381 (32.0)	165 (27.8)	
Unknown	693	279	414	

Cancer stage: according to American Joint Committee on Cancer (AJCC) 8th Edition. Statistically significant values are indicated in bold.HER2: human epidermal growth factor 2. * Receptor status: considered as missing if ≥1 receptor (ER, PR, HER2) missing. ** Positive family history defined as any first or second degree relative affected by breast cancer.

**Table 2 cancers-14-01328-t002:** Quality-of-care indicators stratified by linguistic-geographic region.

Clinical Management	Latin Region (*n* = 1469)			German Region (*n* = 1008)			*p*-Value
	1 Point *n* (%)	No Point *n* (%)	Missing *n*	1 Point *n* (%)	No Point *n* (%)	Missing *n*	
**Pretreatment diagnostic biopsy/FNA** (*n* = 2477)	1464 (99.9)	1 (0.1)	4	1008 (100)	-	-	0.407
**Time to start of treatment <6 weeks** (*n* = 2477)	1289 (93.5)	89 (6.5)	91	931 (95.9)	40 (4.1)	37	**0.014**
**Pathology report indicating ER%, PR%, tumor size and tumor differentiation** (*n* = 2477)	1330 (90.5) All features present	139 (9.5) 1–3 features missing	-	927 (92.0) All features present	81 (8.0) 1–3 features missing	-	0.220
**Locoregional therapy**							
**“No ink on tumor” on final margin** (*n* = 2477)	1042 (80.3)	256 (19.7)	171	772 (84.5)	142 (15.5)	94	**0.012**
**Only 1 surgical procedure** (*n* = 2477)	1004 (70.9)	412 (29.1)	53	716 (72.5)	271 (27.5)	21	0.381
**Removal of > 10 LN when undergoing ALND** (*n* = 1507) Not applicable (*n* = 771) Missing (*n* = 199)	689 (75.8)	220 (24.2)	8	485 (82.5)	103 (17.5)	2	**0.002**
**RT following BCS** (*n* = 1430) Not applicable (*n* = 974) Missing (*n* = 73)	739 (92.5)	60 (7.5)	-	473 (75.0)	158 (25.0)	-	**<0.0001**
**RT boost on tumor bed if BCS** (*n* = 1430) Not applicable (*n* = 974) Missing (*n* = 73)	659 (95.8)	29 (4.2)	111	318 (88.6)	41 (11.4)	272	**<0.0001**
**Post Mastectomy RT** (*n* = 343) Not applicable (*n* = 2049) Missing (*n* = 85)	140 (68.6)	64 (31.4)	-	97 (69.8)	42 (30.2)	-	0.820
**Systemic therapy**							
**Adjuvant chemotherapy if appropriate** (*n* = 1190) Not applicable (*n* = 1004) Missing (*n* = 283)	648 (92.8)	50 (7.2)	-	364 (74.0)	128 (26.0)	-	**<0.0001**
**Adjuvant anti-HER2 therapy if HER2+** (*n* = 571) Not applicable (*n* = 1675) Missing (*n* = 231)	220 (96.1)	9 (3.9)	130	107 (89.2)	13 (10.8)	92	**0.012**
**Endocrine therapy prescribed** (*n* = 1670) Not applicable (*n* = 707) Missing (*n* = 100)	855 (84.5)	157 (15.5)	-	431 (65.5)	227 (34.5)	-	**<0.0001**

Frequency (row %). Statistically significant values are indicated in bold. ALND, axillary lymph node dissection; BCS, breast-conserving surgery; FNA, fine needle aspiration; ER, estrogen receptor; HER2, human epidermal growth factor 2; LN, lymph node; RT, radiation therapy; PR, progesterone receptor; RT, radiotherapy.

**Table 3 cancers-14-01328-t003:** Quality-of-care scores distribution by linguistic-geographic region.

Score	Latin Region		German Region	
	*n* = 1469	%	*n* = 1008	%
100%	554	37.71%	334	18.65%
89–99%	194	13.21%	92	9.13%
79–88%	322	21.92%	213	21.13%
68–78%	191	13.00%	181	17.96%
<67%	208	14.16%	188	18.65%

**Table 4 cancers-14-01328-t004:** Univariate and multivariable associations between clinicopathological factors and overall survival.

	Univariable		Multivariable *	
	Hazard Ratio (95% CI)	*p*	Hazard Ratio (95% CI)	*p*
**Linguistic-geographic region**				
German region	Ref		Ref	
Latin region	0.72 (0.59–0.89)	**0.003**	0.84 (0.68–1.03)	0.097
**Age group**				
≤35	Ref		Ref	
>35	0.79 (0.63–0.98)	**0.031**	0.87 (0.7–1.09)	0.228
**Quality-of-care score**				
1st tertile	Ref			
2nd tertile	0.84 (0.61–1.16)	0.280		
3rd tertile	0.94 (0.70–1.26)	0.670		
100%	0.83 (0.64–1.08)	0.168		
**Period of diagnosis**				
2000–2004	Ref		Ref	
2005–2009	0.68 (0.54–0.87)	**0.002**	0.59 (0.46–0.76)	**<0.0001**
2010–2014	0.66 (0.48–0.89)	**0.008**	0.59 (0.43–0.81)	**0.001**
**Tumor differentiation**				
Well	Ref		Ref	
Moderately	3.11 (1.58–6.12)	**<0.0001**	2.36 (1.19–4.67)	**0.014**
Poorly	5.22 (2.68–10.17)	**<0.0001**	3.20 (1.59–6.42)	**0.001**
Missing	4.72 (2.09–10.69)	**<0.0001**	2.90 (1.24–6.76)	**0.014**
**Tumor stage**				
I	Ref		Ref	
II	2.75 (1.97–3.83)	**<0.0001**	2.42 (1.73–3.38)	**<0.0001**
III	6.34 (4.46–9.02)	**<0.0001**	5.55 (3.98–7.93)	**<0.0001**
Missing	4.91 (3.18–7.58)	**<0.0001**	4.32 (2.77–6.75)	**<0.0001**
**Tumor subtype**				
Triple negative	Ref			
Luminal A like	0.41 (0.28–0.58)	**<0.0001**	0.62 (0.41–0.95)	**0.028**
Luminal B like	0.77 (0.59–1.02)	0.067	0.83 (0.63–1.11)	0.213
HER2+	1.12 (0.73–1.73)	0.598	0.95 (0.61–1.47)	0.810
Unknown	0.67 (0.48–0.94)	**0.019**	0.66 (0.46–0.96)	**0.031**

Statistically significant values are indicated in bold. Ref: reference category. ***** Adjusted for linguistic-geographic region, age, period of diagnosis, tumor differentiation, stage, and subtype.

## Data Availability

E.R. had full access to the data and takes responsibility for the integrity of the data and the accuracy of the data analysis.

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
