# Peer review of "Management and Outcome of Young Women (≤40 Years) with Breast Cancer in Switzerland"

_cancers, 2022, doi:10.3390/cancers14051328_

Round 1
Reviewer 1 Report
Dear authors,
I have valued the opportunity to review your paper of your epidemiological analysis of breast cancer survival in young women diagnosed with breast cancer in two geographic areas/linguistic populations of Switzerland. The paper appears to have two aims; to describe the disparity between two population groups and to identify factors that may cause this disparity.
My comments:
- Aim - there is a need to be clear in the intentions of your paper regarding descriptive vs causal inference epidemiology (and prediction)
- A re-structure would provide more clarity, as well. The introduction suggests the primary aim is to investigate difference in treatment and treatment outcomes (survival is not mentioned as an outcome). The structure of the methods places emphasis on the Quality of Care score (as well as the tumour sub-types), and did not describe treatment variables. ‘Treatment outcome’ variables were not described either, until the end of the methods in the Statistical Analysis section, which made it clear that breast cancer survival (using relative survival methods) was the primary outcome. Why the Introduction and Title suggest region is the primary ‘explanatory’ variable, the results seem to focus more on quality of care (often discussed first).
- I recommend that the Introduction be revised and expanded on. If the primary outcome is relative survival, I recommend providing the national survival statistics for breast cancer, and specifically for young women with breast cancer. If the primary purpose is to describe disparities between regions, then perhaps give statistics to support the hypothesis that breast cancer survival in young women is likely to be differential in these two groups/regions. Likewise, if you are trying to determine causal factors of this disparity you need to background this, including introducing the reader to your main co-variates (quality of care, cancer stage/grade and time trends).
- The Method section does not currently provide sufficient information to critique the study or replicate it. I think again, a clear aim will help you do this. A restructure may look like: 2.1 Study Population, 2.2 Data Sources and Variables, 2.3 Co-variate Measures, 2.4 Outcome Measure (survival time), 2.5 Statistical Analysis, and 2.6 Ethics and Governance. Governance may not be as critical, I am not familiar with the Switzerland context. If in Switzerland, there are socio-political power imbalances between the two regions, it would also be important to ensure that the marginalised region has had a ‘voice’ in the research (ideally in all components), and have governance of their data.
- Define 'not routinely collected' – to me it has always meant that the register may have the variable but that the data is incomplete (missing, possibly inaccurate, possible inconsistently coded). As such, these variables are typically not available to researchers. So I assumed you did not have these variables, though later I realised that you did (I think).
- Tumour sub-types, family history and quality of care score are the only variables described in detail, and not even in the same section. All variables – how they are defined, measured, categorised, etc – should be described.
- It is not clear until the statistical analysis section what your outcome is. There should be a section that defines survival time as your primary outcome. This should tell us about time zero (what defines cohort entry and the commencement of follow-up time) and when follow-up time ends (time of death - what about end of study? Is there any other reason why someone would be censored (e.g. migration, diagnosis of secondary or recurrent cancer))
- In Co-variate Measures, I recommend you first define how region was measured. E.g. Was is defined using one or multiple variables in the register, where did the register gain information on these variables from, are the variables complete and accurate, can people change group membership overtime or is it assigned at birth?
- In the Co-variate Measures section, I think you should then describe the other variables you include in the multivariable analysis: Quality-of-Care Score, tumour stage, grade, and year. How are they defined/created and how are they classified? Did you consider age? If not, why not? It is commonly an important confounder, but is there a reason to not suspect that in this case?
- Quality of care score - were clinical experts used in the determination/confirmation of the algorithms developed to code the quality of care scores according to the clinical guidelines?
- Leading on from the last question, was the data used to derive the quality of code score 'clean' or 'messy' - in other words, was there any need for a level of intepretation of the data to determine coding of the score? If so, were clinician experts involved in this?
- What do you mean that if the quality of care score was under 100% you classified them into groups? Did someone women have a quality score higher than 100%? How was this possible and what happened to them?
- There is a mention that missing datapoints were excluded. This is an important potential source of bias. Are you able to quantify how much data is missing (by variable and by participant). Is there a point at which you would consider a variable or a person to have too much missing information? You perform a sensitivity analysis (as per the results), and the method of this needs to be described in the method section of the manuscript.
- Did you investigate any potential interactions or effect modification caused by cofounders between the primary relationships of interest (region and OS survival and region and breast cancer survival).
- How were variables selected into the model? This needs to be described and what is appropriate this will depend on the study aim and, therefore, the purpose of the multivariable analysis (descriptive, causal inference, predictive). As you refer to the factors at the end of the results as ‘predictors’, I want to say that I do not think this is predictive analyses and therefore I would not refer to the variables as predictors. To me it sounds like the aim of the multivariable analysis is causal inference. You have included region, year, quality of care score, and tumour stage/grade into one model. If we think about region as the exposure (just for this purpose) and survival as the outcome, then quality of care and tumour stage and grade which occurring after study entry (cancer diagnosis) and if influenced by region would instead be on the causal pathway, are more likely to be mediators. So, it may be more informative to understand how much of the survival disparity is mediated through differences in quality of care, differences in stage/grade of cancer at diagnosis. Unless, of course, you do not believe these factors to be on the causal pathway? Alternatively, you may want to see whether there is evidence that quality of care may explain survival, even beyond any other reasons associated with being from a particular region. You could then create a model with quality of care score as the exposure; region, year and cancer stage/grade as confounders; and survival as the outcome. The confounders occur before the exposure and it would be reasonable to assume that they may each explain both exposure (quality of care) and outcome (survival), i.e. confounding variables. Another model could then be created with cancer stage in a similar way, but I would argue that quality of care is likely a mediator between cancer stage—survival, and thus should not be adjusted for (as there would then be risk it could create a spurious relationship between cancer stage and survival). Another alternative is to conduct the two models as just described, but stratified by region (rather than include region in the multivariable model).
- There are other variables that I would expect to be included and that I would have thought would have been included in the registry, such as age, comorbidities prior to cancer diagnosis, socioeconomic status, area-level remoteness/deprivation, etc. Some explanation for the exclusion of such variables would be useful.
- Table 2 - The title should say Quality of Care Indicators, rather than scores, as this table does not contain the scores, but rather the series of indicators (yes;1/no;0) that are used to derive the score.
- In addition to my comment about variable selection (q15), the lack of data on other likely confounders - other clinical factors (e.g. comorbidities), other demographics (e.g. socioeconomic status), and socio-political factors (e.g. discrimination) - is another major potential source of bias that should be considered in your discussion.
Author Response
Rev #1 Dear authors, I have valued the opportunity to review your paper of your epidemiological analysis of breast cancer survival in young women diagnosed with breast cancer in two geographic areas/linguistic populations of Switzerland. The paper appears to have two aims; to describe the disparity between two population groups and to identify factors that may cause this disparity.
My comments:
- Aim - there is a need to be clear in the intentions of your paper regarding descriptive vs causal inference epidemiology (and prediction)
RESPONSE: This was a descriptive analysis and therefore only associations were evaluated and not causal inference. The abstract and the introduction have been modified to better define the study aims (background/methods of the abstract, page 11 and the last paragraph of the introduction, page 12).
- A re-structure would provide more clarity, as well. The introduction suggests the primary aim is to investigate difference in treatment and treatment outcomes (survival is not mentioned as an outcome).
RESPONSE: We have clarified in the introduction that the aims of this study were to evaluate quality of care and survival (last paragraph of the introduction, page 12).
The structure of the methods places emphasis on the Quality of Care score (as well as the tumour sub-types), and did not describe treatment variables. ‘Treatment outcome’ variables were not described either, until the end of the methods in the Statistical Analysis section, which made it clear that breast cancer survival (using relative survival methods) was the primary outcome.
RESPONSE: Breast cancer survival was not the primary aim of this study. The primary aim was to evaluate quality of care. We have followed your advice and restructured the methods section (page 12-14). Treatment outcomes are described in the quality-of-care paragraph (page 13).
Why the Introduction and Title suggest region is the primary ‘explanatory’ variable, the results seem to focus more on quality of care (often discussed first).
RESPONSE: The current title (Management and Outcome of Young Women (≤ 40 years) with Breast Cancer in Switzerland) emphasizes the main aims of the study: the first was evaluation of management (i.e. quality of care) and the second was evaluation of outcome (i.e. survival). The linguistic/geographic region was not the primary explanatory variable and it is not mentioned in the title. We have changed the abstract and the introduction accordingly (background/methods of the abstract, page 11 and the last paragraph of the introduction, page 12).
- I recommend that the Introduction be revised and expanded on. If the primary outcome is relative survival, I recommend providing the national survival statistics for breast cancer, and specifically for young women with breast cancer.
RESPONSE: The primary aim of the study was quality of care and not survival. However, since survival was a secondary aim, we have added national survival statistics in the introduction (first paragraph of the introduction, page 12).
If the primary purpose is to describe disparities between regions, then perhaps give statistics to support the hypothesis that breast cancer survival in young women is likely to be differential in these two groups/regions.
RESPONSE: Data on disparities in breast cancer treatment and survival in Switzerland have been previously reported (Ref. 12 and 13). However, data on treatment disparities on young women are lacking, which is what prompted us to design this study. This has been now added in the introduction.
Likewise, if you are trying to determine causal factors of this disparity you need to background this, including introducing the reader to your main co-variates (quality of care, cancer stage/grade and time trends).
RESPONSE: Determination of causal factors for disparities in treatment and outcome of breast cancer in Switzerland was not among our aims. Our study was descriptive.
- The Method section does not currently provide sufficient information to critique the study or replicate it. I think again, a clear aim will help you do this. A restructure may look like: 2.1 Study Population, 2.2 Data Sources and Variables, 2.3 Co-variate Measures, 2.4 Outcome Measure (survival time), 2.5 Statistical Analysis, and 2.6 Ethics and Governance.
RESPONSE: We appreciate your help to improve of our manuscript. The methods section has been restructured as you suggested (page 12-14).
Governance may not be as critical, I am not familiar with the Switzerland context. If in Switzerland, there are socio-political power imbalances between the two regions, it would also be important to ensure that the marginalised region has had a ‘voice’ in the research (ideally in all components), and have governance of their data.
RESPONSE: Thank you for raising this important point. All three linguistic/geographic regions were included in the study and after ethical approval was obtained each single registry collected data for their own region.
- Define 'not routinely collected' – to me it has always meant that the register may have the variable but that the data is incomplete (missing, possibly inaccurate, possible inconsistently coded). As such, these variables are typically not available to researchers. So I assumed you did not have these variables, though later I realised that you did (I think).
RESPONSE: Thank you for raising this other important point. Some clinicopathological (i.e. BMI) and treatment characteristics (i.e tumorboard discussion) are not recorded in the registry databases. Therefore, an ad-hoc questionnaire was created and sent to each registry where dedicated registrars reviewed each patient chart and extracted these data when available. This has been clarified in the methods section (Variable paragraph, page 13).
- Tumour sub-types, family history and quality of care score are the only variables described in detail, and not even in the same section. All variables – how they are defined, measured, categorised, etc – should be described.
RESPONSE: Thank you, description of all variables routinely collected by the registries has now been added to the method section (page 12).
- It is not clear until the statistical analysis section what your outcome is. There should be a section that defines survival time as your primary outcome.
RESPONSE: Thank you, the method section has been revised and we added a paragraph describing survival outcomes (outcome measure paragraph, page 14).
This should tell us about time zero (what defines cohort entry and the commencement of follow-up time) and when follow-up time ends (time of death - what about end of study? Is there any other reason why someone would be censored (e.g. migration, diagnosis of secondary or recurrent cancer)
RESPONSE: Thank you. This has been added in the outcome measure paragraph (page 14).
- In Co-variate Measures, I recommend you first define how region was measured. E.g. Was is defined using one or multiple variables in the register, where did the register gain information on these variables from, are the variables complete and accurate, can people change group membership overtime or is it assigned at birth?
RESPONSE: Thank you. The attribution to the linguistic/geographical region of a patient was determined based on the residence of the patient at the time of diagnosis. This is unrelated to the place of birth or nationality. If the patient changed canton she was lost at follow-up on the day of migration. This has been added to the methods (Linguistic/geographic region paragraph, page 14).
- In the Co-variate Measures section, I think you should then describe the other variables you include in the multivariable analysis: Quality-of-Care Score, tumour stage, grade, and year. How are they defined/created and how are they classified? Did you consider age? If not, why not? It is commonly an important confounder, but is there a reason to not suspect that in this case?
RESPONSE: Thank you. All covariates are now listed in the methods section (page 13). All variables significant in the univariate analysis were included in multivariable analysis, these included the linguistic/geographic region, age, period of diagnosis, tumor differentiation, stage and subtype. This has now been clarified in the methods (page 14). We realized that age was erroneously not mentioned in the text (methods) and in the footnote of table 4, however age was always included in our model. This has now been corrected.
- Quality of care score - were clinical experts used in the determination/confirmation of the algorithms developed to code the quality of care scores according to the clinical guidelines?
RESPONSE: Yes, many of the study investigators are clinicians with dedicated practice to breast cancer care and they were actively involved in the determination of the quality-of-care score.
- Leading on from the last question, was the data used to derive the quality of code score 'clean' or 'messy' - in other words, was there any need for a level of intepretation of the data to determine coding of the score? If so, were clinician experts involved in this?
RESPONSE: Data collection was standardized and digitalized to avoid misinterpretation. The data was “clean” and not subject to any interpretation.
- What do you mean that if the quality of care score was under 100% you classified them into groups? Did someone women have a quality score higher than 100%? How was this possible and what happened to them?
RESPONSE: We categorized the quality of care in 4 categories: patients with a score of 100% were included in one group, the others were divided in tertiles. There were no patients with a score >100%. This has been clarified in the quality-of-care score section, page 14.
- There is a mention that missing datapoints were excluded. This is an important potential source of bias. Are you able to quantify how much data is missing (by variable and by participant). Is there a point at which you would consider a variable or a person to have too much missing information? You perform a sensitivity analysis (as per the results), and the method of this needs to be described in the method section of the manuscript.
RESPONSE: Before data analysis began, missingness was quantified and only variables with ≤ 30% missingness were included in the score. However, no individuals were excluded as the score was based on non-missing indicators only (i.e. if a patient had 3 missing indicators, the score was based on the 9 remaining ones that were available).
We performed two sensitivity analyses considering extreme case scenarios. This has been added to the methods (statistical analysis section, page 14).
- Did you investigate any potential interactions or effect modification caused by cofounders between the primary relationships of interest (region and OS survival and region and breast cancer survival).
RESPONSE: Our primary relationship of interest was not between region and survival, but rather between quality of care and survival. Based on prior reports, we hypothesized that differences in quality of care across Swiss regions would account for differences in outcome, however, our analyses showed that this is not the case. Patients treated in the Latin and German regions had different tumor characteristics (stage, tumor subtype and tumor grade) and after adjustment for these variables the association of region with survival was no longer significant (Table 4). The baseline tumor characteristics difference between the two regions is most likely due to random variation as our study only covered 45% of the Swiss population. This has now been clarified in the discussion (page 17). We did not think that an interaction analysis would have given neither biologically nor clinically meaningful results, because there is no biological or clinical reason to believe that tumor biology would differ across the country.
- How were variables selected into the model? This needs to be described and what is appropriate this will depend on the study aim and, therefore, the purpose of the multivariable analysis (descriptive, causal inference, predictive).
Variables were a priori selected based on clinical knowledge (i.e. tumor stage, subtype, grade, period of diagnosis and age) and based on the aims of the study (Quality of care score and linguistic/geographic region). This has now been described in the covariates paragraph, page 13).
As you refer to the factors at the end of the results as ‘predictors’, I want to say that I do not think this is predictive analyses and therefore I would not refer to the variables as predictors.
We agree with you and we have changed this throughout the manuscript.
To me it sounds like the aim of the multivariable analysis is causal inference. You have included region, year, quality of care score, and tumour stage/grade into one model. If we think about region as the exposure (just for this purpose) and survival as the outcome, then quality of care and tumour stage and grade which occurring after study entry (cancer diagnosis) and if influenced by region would instead be on the causal pathway, are more likely to be mediators. So, it may be more informative to understand how much of the survival disparity is mediated through differences in quality of care, differences in stage/grade of cancer at diagnosis. Unless, of course, you do not believe these factors to be on the causal pathway? Alternatively, you may want to see whether there is evidence that quality of care may explain survival, even beyond any other reasons associated with being from a particular region.
You could then create a model with quality of care score as the exposure; region, year and cancer stage/grade as confounders; and survival as the outcome. The confounders occur before the exposure and it would be reasonable to assume that they may each explain both exposure (quality of care) and outcome (survival), i.e. confounding variables. Another model could then be created with cancer stage in a similar way, but I would argue that quality of care is likely a mediator between cancer stage—survival, and thus should not be adjusted for (as there would then be risk it could create a spurious relationship between cancer stage and survival). Another alternative is to conduct the two models as just described, but stratified by region (rather than include region in the multivariable model).
RESPONSE: As mentioned above we did not aim at assessing causal inference, however we followed your advice and did a mediation analysis (considering region as the exposure, survival as the outcome and stage, grade and quality-of-care as mediators). The association between region and survival is mostly mediated by biological factors such as tumor grade (24% at 5 years after diagnosis) and by stage (29.4% at 5 years) and not by quality-of-care score (4.4% at 5 years). These results reinforce the multivariable cox model results which did not include quality of care (as not significant in UVA) and did not show a significant association between region and survival but only between tumor characteristics and survival.
- There are other variables that I would expect to be included and that I would have thought would have been included in the registry, such as age, comorbidities prior to cancer diagnosis, socioeconomic status, area-level remoteness/deprivation, etc. Some explanation for the exclusion of such variables would be useful.
RESPONSE: Clinical variables known to be associated with BC survival were a priori selected and age was included in our analysis. Given that this was a cohort of young women (<40 years), we did not expect that adjusting for comorbidities would have impacted our results. Although important other sociodemographic variables were not available to us. It should be noted that the Swiss healthcare system is characterized by universal coverage ensured through mandatory health insurance, and therefore socioeconomic disparities are less pronounced than in other regions.
- Table 2 - The title should say Quality of Care Indicators, rather than scores, as this table does not contain the scores, but rather the series of indicators (yes;1/no;0) that are used to derive the score.
RESPONSE: Thank you, this has now been corrected.
- In addition to my comment about variable selection (q15), the lack of data on other likely confounders - other clinical factors (e.g. comorbidities), other demographics (e.g. socioeconomic status), and socio-political factors (e.g. discrimination) - is another major potential source of bias that should be considered in your discussion.
RESPONSE: In the discussion (limitation section, page 17) we added a sentence about the lack of information about other potential confounders.
Reviewer 2 Report
This paper looks at breast cancer in young women (<40 years old) in Switzerland, finding a difference in quality of care and survival rates between 2 cohorts. Thank you for conducting this important work, I find the topic of young women BC very interesting.
Intro is brief, concise and relevant to the topic at hand. Line 62-63: Word missing between "mostly" and "responsibility".
Methods - why was 2014 chosen as the cutoff date for collection? Is it possible to get more recent data, as this is now almost 10 years old. I am curious about why these 2 regions were chosen at the exclusion of others? Was this simply a data collection issue (not being able to reach the other areas) or was this done deliberately?
Results are clearly presented, with correct analyses performed to answer research questions of interest. Discussion is also well presented, with clear strengths and limitations presented. Conclusions are appropriate and don't amplify the findings of the study.
Line 235: "the mean quality of care score" - care is missing.
Author Response
Rev #2: Comments and Suggestions for Authors
- This paper looks at breast cancer in young women (<40 years old) in Switzerland, finding a difference in quality of care and survival rates between 2 cohorts. Thank you for conducting this important work, I find the topic of young women BC very interesting.
RESPONSE: Thank you, we appreciate your positive feedback.
- Intro is brief, concise and relevant to the topic at hand. Line 62-63: Word missing between "mostly" and "responsibility".
RESPONSE: This has now been corrected (page 12).
- Methods - why was 2014 chosen as the cutoff date for collection?
RESPONSE: The study was designed and conceived in 2016. At that time completed data were available up to 12/2014.
- Is it possible to get more recent data, as this is now almost 10 years old.
RESPONSE: To collect new incident cases, we would need to initialize a new study. For the cohort included in this study, data not routinely collected by the cancer registries were collected between 2017-2018 and follow-up ended in 12/2018. Survival data at national level are now available up to 12/2019, however extending the follow-up period would only add 1 year of follow up to the present cohort. An update analysis is planned to take place at the end of 2022 when survival data will be available up to 2020. To capture recent changes in the management of early BC (such as the use of post-NAC Capecitabine or TDM-1 in triple negative and HER2+ breast cancer with residual disease post neoadjuvant chemotherapy or the use of immunotherapy in the neoadjuvant setting) we would need to include patients treated from 2018 onwards, for which the median follow-up up would be too short.
- I am curious about why these 2 regions were chosen at the exclusion of others? Was this simply a data collection issue (not being able to reach the other areas) or was this done deliberately?
RESPONSE: The 26 Swiss cantons are divided in 3 linguistic regions (German, French and Italian). We grouped the French and the Italian speaking regions together (Latin region) and compared them to the German region, as previously done in other Swiss studies, because previous reports have shown differences in cancer risk avoidance behaviors, in the use of health care services and in socioeconomic status in these two regions. In 2016, only 10 registries had data covering the study period. We were able to include nine of these ten registries. Unfortunately, one registry in the German speaking part of the country was not able to participate due to financial constraints.
- Results are clearly presented, with correct analyses performed to answer research questions of interest. Discussion is also well presented, with clear strengths and limitations presented. Conclusions are appropriate and don't amplify the findings of the study.
RESPONSE: Thank you, we appreciate your positive feedback.
- Line 235: "the mean quality of carescore" - care is missing.
RESPONSE: Thank you, this has been corrected (page 16).
Round 2
Reviewer 1 Report
Dear authors,
Thank you for your considered responses. I feel the edits have give your manuscript clarity and it is now much clearer what you intended to do and how you did it. I enjoyed reading it and I have no further suggestions.
Thank you